# DAC^®^_,_ a Hyaluronan Derivative in the Form of a Gel, Is Effective in Preventing Periprosthetic Joint Infection During Arthroplasty Revision in Patients with Comorbidities: A Retrospective, Observational, 1:1-Matched Case–Control Clinical Investigation

**DOI:** 10.3390/biomedicines13102408

**Published:** 2025-09-30

**Authors:** Giuseppe Ricciardi, Giancarlo Giuliani, Arminio Armando, Raffaele Quitadamo, Rosario Demita, Costantino Stigliani

**Affiliations:** Orthopedic and Traumatology Division, Teresa Masselli Mascia Hospital, 71016 San Severo, Foggia, Italy; giancarlo.giuliani@aslfg.it (G.G.); arminio.armando@aslfg.it (A.A.); raffaele.quitadamo@aslfg.it (R.Q.); rosario.demita@aslfg.it (R.D.); costantino.stigliani@aslfg.it (C.S.)

**Keywords:** revision arthroplasty, periprosthetic joint infection, hydrogel, absorbable barrier

## Abstract

**Background/Objectives**: Joint arthroplasty revision and comorbidities are considered two increased risk factors for periprosthetic joint infection (PJI), a complication that may lead to prolonged hospital stay, continued antibiotic therapy, and serious consequences, including amputation and, in extreme cases, death of the patient. DAC^®^ is an absorbable barrier in the form of a gel that, when applied as a coating, protects implants from bacterial colonization. The aim of this case–control study was to explore whether the device could decrease the risk of PJI in a cohort of patients who underwent arthroplasty revision and were affected by comorbidities. **Methods**: We carried out a retrospective 1:1-matched case–control investigation in 96 patients who underwent arthroplasty revision between January 2023 and December 2024; these patients had at least 6 months of follow-up, had comorbidities, and were treated with DAC^®^ gel. The control group consisted of 96 subjects who received standard of care. Demographics, comorbidities, type of arthroplasty, adverse event onset, and incidence of PJI were recorded for all patients. **Results**: No significant differences in relevant demographics, type of arthroplasty revision, or number or type of comorbidities, except for smoking, were observed between the two groups. At 6-month follow-up, no PJIs were recorded in the DAC^®^ treatment group, whereas five (5.2%) PJIs were observed in the control group (*p* = 0.0235). No adverse event or impairment of implant osseointegration related to the use of DAC^®^ was observed. **Conclusions**: The DAC^®^ bioabsorbable hydrogel acts as a physical barrier when applied over an arthroplasty revision implant, protecting it from bacterial adhesion and preventing biofilm formation.

## 1. Introduction

Joint replacement, involving either the knee or hip, has become one of the most common surgical procedures and significantly increases the quality of life of patients, alleviating the debilitating hip or knee pain caused by degenerative disease or trauma [1]. However, while there have been advances in both surgical techniques and medical treatment strategies over the years, the risk of periprosthetic joint infections (PJIs) has increased in incidence and severity, particularly in more complex patients, such as those affected by comorbidities or specific associated disease or who need revision surgery [2,3,4]. Furthermore, some clinical investigations reported that the incidence of PJI, specifically in hip arthroplasty, is underestimated, and the conclusions highlight the crucial need for standardized diagnostic criteria and monitoring methods to accurately identify and path cases of PJI [5]. The rate of Surgical Site Infection (SSI) in knee and hip arthroplasty varies between 0.5 and 2.3% with a range of 0–2.3% for SSI and 0–1.7% for deep or organ/space SSI, as reported in the Annual Epidemiological Report of the European Centre for Disease Prevention and Control (ECDC) [6] and in the literature. However, incidences as high as 4% in cases of revision or in patients who are affected by comorbidities such as obesity, smoking, and diabetes have been observed [7,8,9].

Despite the relatively low incidence of PJI, the financial burden for the health system remains high [10]. The literature reports that the overall cost of total knee arthroplasty (TKA) infection treatment in the U.S. increased from USD 206.1 million in 2002 to USD 518.2 million in 2017, whereas the total cost of total hip arthroplasty (THA) infection increased from USD 166.6 million to USD 384 million during the same period. Using Poisson regression models, the authors projected the expected number of cases and costs of both THA and TKA with PJI, and they reported that by 2030, the estimated total national hospital cost will reach USD 1.85 billion [11]. Furthermore, the management of PJI requires significantly more resources, including administrative work, than that of primary total joint arthroplasty [12].

The reported successful control of infection after the surgical management of patients with two-stage arthroplasty revision is quite variable, with success rates ranging from 59% to 100% and a good success frequency in the treatment at mid-term to long-term follow-up; however, polymicrobial and methicillin-resistant *Staphylococcus aureus* (MRSA) infections are poor prognostic factors, making the eradication of infection more difficult [13,14], leading to divergent opinions and discussions with respect to the best approach for treating and managing PJI in revision surgery [15,16]. Systemic antibiotic administration displayed proven efficacy in decreasing the risk of infection associated with the use of foreign bodies such as prostheses and osteosynthesis devices [17]; however, identifying antibiotic best practices in the fight against PJI is limited by a lack of evidence [18]. The routine topical use of antibiotics delivered at joint districts, including soft tissues or intrawound locations, seems to lead to satisfactory clinical results in reducing PJIs [19]; nevertheless, some authors doubt the clinical effectiveness of this approach [20].

The mechanism of PJIs is well known: bacteria in the early phase of surgery colonize and adhere to the surface of an implant, forming biofilms in a polysaccharide-based matrix that protects them from the antimicrobial activity of systemic antibiotic prophylaxis and the immune system.

The physico-chemical surface properties of an arthroplasty titanium implant, including topography and chemistry, are important in the promotion or inhibition of cell and bacterial adhesion. Gasik M et al. [21] described how a hydrophobic surface might be a suitable substrate for bacterial colonization, while a hydrophilic surface can prevent bacterial adhesions and biofilm formation.

For instance, *S. aureus* is identified as the most common pathogen responsible to majority of the implant related infections, and it typically is hydrophilic in nature due to the presence of highly negative charged teichoic acids as cell wall constituents [22], preferring attachment to a hydrophobic surface [23]; hence, a hydrophilic surface can inhibit the adhesion and maturation of biofilm.

Preventing PJIs is therefore a key factor in avoiding biofilm formation because when biofilm formation occurs, infection will persist until the surgical removal of infected implants and dead tissue [24]. Long-term antibiotic administration and wound debridement with implant retention (DAIR) may achieve high rates of infection control and should be considered an option for patients who are unfit for major surgery; however, the literature reports conflicting clinical outcomes [25]. As mentioned before, the local delivery of antibiotics is considered an effective method for both preventing and treating infection [26]; in 1981, Buchholz et al. [27] introduced the use of gentamycin-loaded bone cement in total hip and knee surgery for the prevention of infection. Although sound evidence is lacking, local release of gentamycin might prevent infection in cemented total joint arthroplasty. Cement spacers loaded with one or more antibiotics, usually gentamicin and vancomycin at high doses (up to 4 g per 40 g of cement), are recommended for two-stage revision arthroplasty to eradicate infection [28].

While two-stage revision remains a successful treatment decision that is deemed the gold standard, such a procedure can trigger discomfort and a high emotional cost, therefore resulting in a relatively negative impact on the quality of life of patients and their families. Moreover, multiple operations and prolonged hospital stays impose a significant burden on National Health Systems [29,30]. Furthermore, in long-term follow-up of more than 5 years, this technique resulted in relatively high infection eradication; however, high mortality and antibiotic-resistant organisms were a greater risk detected, as reported in a multicenter retrospective clinical review [31].

Prevention of PJI is imperative, and over the last forty years, several modalities have been encouraged. Although the administration of i.v. antibiotics such as cephazolin perioperatively was the milestone in the prevention of infection in orthopedics, ultraclean operating rooms, reduced operative times, and the use of irrigation systems are also considered relevant factors, and, as mentioned above, in cemented joint replacements, the addition of gentamicin to PMMA cement has also been used to prevent infection [32]. In conclusion, strategies for reducing the risk of PJI involve preoperative, intraoperative, or postoperative stages. Preoperative measures are important because they are the first line of defense, and preoperative clinical evaluation allows for a key step in screening and diagnosing underlying comorbidities and optimizing modifiable risk factors before elective surgeries. These modifiable factors include autoimmune disease, obesity, diabetes mellitus, smoking, alcohol abuse, and a history of steroid administration [33].

Local prophylactic antibiotic therapy using carriers based on inorganic beads may represent a valid way to prevent PJI infection after the administration of systemic antibiotics, as reported in a randomized clinical trial [34]. Alternatively, biopolymeric hydrogel surface coatings acting as a physical barrier to avoid bacterial adhesion and delivering antibiotics in the early phase of surgery may represent an effective approach for preventing PJIs [35,36,37]. The primary function of hydrogels with hydrophilic properties is to create a temporary physical barrier to avoid the adhesion of planktonic bacteria, making the underlying hydrophobic implant surface unavailable and thereby inhibiting bacterial colonization on the implant and subsequent biofilm formation [21,38].

The DAC^®^ (Defensive Antibacterial Coating) is a biocompatible device based on hyaluronan (HY) and poly-D,L-lactic acid (PDLLA); it has demonstrated the ability to provide a physical barrier against bacterial adhesion [39] and, when applied even in combination with antibiotics, has shown proven efficacy in arthroplasty and traumatology [40,41]. Therefore, according to the instructions for use, the purpose of this case–control investigation was to analyze and compare the incidence of PJIs after the application of DAC^®^ as a physical barrier without loading it with antibiotics in knee or hip uncemented arthroplasty revision in a cohort of patients affected by comorbidities that could have posed an additional risk for PJI onset.

## 2. Materials and Methods

### 2.1. Study Design and Patient Enrollment

We retrospectively collected data from the internal clinical records of our Institution to identify patients who had undergone hip or uncemented knee arthroplasty with either septic or aseptic revision from July 2023 to July 2024. Other inclusion criteria were as follows: (a) adult patients; (b) patients who provided signed informed consent before surgery; (c) patients who underwent revision with uncemented implants; (d) patients affected by one or more comorbidities; (e) patients who had at least 6 months of follow-up. Patients who received DAC^®^ hydrogel treatment (Adler Ortho S.p.a., Cormano, Milan, Italy) over the surface of the revision implant were 1:1 matched with a group of patients with the same demographic characteristics treated with conventional therapy (non-DAC^®^ group) from June 2021 to April 2023 before the use of the DAC^®^ device was authorized in our operating unit. This investigation was reported to the local I.R.B. and all patients’ data were collected in a strictly anonymous form to protect any personal identification in compliance with the applicable local laws and privacy regulations (for this retrospective case–control investigation, reference is made to the recent reform/amendment of Art. 110—Provision No. 298/2024—of Italian Legislative Decree No. 196/2003).

### 2.2. DAC^®^ Gel Coating and Standard Treatment

All patients were given preoperative antibiotic prophylaxis, and postoperatively, they received low-molecular-weight heparin for deep vein thrombosis prophylaxis.

The subjects in the DAC^®^ group had arthroplasty revision implants homogeneously coated with the device in the form of gel after hydration of the powder contained in a sterile syringe with sterile water, as shown in Figure 1a–c. A review of medical records and 1:1 case-matching were carried out by an independent data manager who was not involved in patient care. Demographic data related to age, sex, body mass index (BMI), anatomical site of the implant (knee or hip), and the number and type of comorbidities, such as diabetes, hypertension, smoking, obesity, heart disease, neoplasia alcohol abuse, autoimmune disease, and other comorbidities, were recorded, and a preoperative internationally validated risk factor calculator, specifically designed to assess the risk of PJI occurrence, was also employed [42].

### 2.3. Assessment of PJI

The diagnosis of PJI (primary endpoint) was completed by applying the criteria proposed in 2011 by the Musculoskeletal Infection Society and then modified in 2013 from the International Consensus Meeting held in Philadelphia, published in 2018 [43], and the conventional criteria established by the Guidelines of the Centers for Disease Control and Prevention (CDC) for the prevention of SSI [44]. Diagnosis of PJI was recognized when the following major criteria were met: (a) two positive growths of the same organism using standard culture methods and (b) a sinus tract communicating with the joint. The samples underwent aerobic and anaerobic cultures with a 14-day incubation period to detect slow-growing organisms.

Follow-up visits for the primary endpoint were considered those at 3 and 6 months according to the guidelines and literature on PJI evaluation: early PJIs occur within the initial 4 weeks following the primary arthroplasty. Early PJIs are typically caused by highly virulent organisms such as *S. aureus*, aerobic Gram-negative bacilli, beta-hemolytic *Streptococci*, and *Enterococcus* spp. Delayed PJIs occur between 3 and 12 months following surgery and are caused by pathogens of slight virulence, including coagulase-negative *staphylococci*, *C. acnes*, and *enterococci*. Late PJIs occur 1 to 2 years after arthroplasty and are mainly hematogenous in nature.

The secondary endpoints were the incidence and severity of adverse events, the rate of revision, and the Rx assessment of implant osseointegration.

### 2.4. Statistical Analysis

All the data were analyzed using IBM SPSS 29 for Windows. Descriptive statistics were presented separately by treatment group and were retrospectively compared. Continuous variables were presented as the median and interquartile range and were compared using the Mann–Whitney U test and Student’s *t* test. Categorical variables were reported as the number of cases and percentages and were compared using the chi-square test and Fisher’s exact test. A two-tailed *p*-value < 0.05 was considered statistically significant.

## 3. Results

### 3.1. Patients’ Baseline Characteristics

Ninety-six patients who had undergone uncemented hip or knee arthroplasty revision by a standard surgical protocol and who were affected by one or more comorbidities and who received DAC^®^ gel intraoperatively were identified and compared with ninety-six matched patient controls who received conventional surgery but did not receive DAC^®^ gel. Both groups underwent follow-up visits 3 and 6 months after surgery.

There were no significant differences in demographic characteristics between the two groups: the median age of the population in the DAC^®^ group (46 F, 50 M) was 63 (IQR 12.5) years, with a median BMI of 25.6 (IQR 3.3) kg/m^2^, whereas the median age of the control group (46 F, 50 M) was 63 (IQR 9) years, with a median BMI of 25.7 (3.1) kg/m^2^.

Each group had an identical distribution of hip and knee arthroplasty revision, while five patients in the DAC^®^ group and four in the control group underwent septic arthroplasty revision. In the DAC^®^ group, 97.9% of patients received cefazoline and 2.1% of patients received vancomycin as antibiotic prophylaxis preoperatively, while in the control group, all patients received i.v. administration of cefazoline. A significantly greater median length (min.) of surgery (*p* = 0.0001) was recorded in the control group (90, IQR 15) than in the DAC^®^ group (80, IQR 20).

The most prevalent comorbidities for the overall population were smoking, obesity, and diabetes. No statistically significant differences were observed in the number and type of comorbidities between the two groups, except for smoking, which was more frequent in the DAC^®^ group (*p* = 0.0224). Likewise, for the calculated risk factor (%), no statistically significant differences were seen between the two groups. The demographic characteristics of both groups are presented in Table 1.

### 3.2. PJIs and Adverse Event Assessment

Concerning the primary endpoint, PJIs (deep/organ space) were observed in five patients belonging to the control group but none in the DAC^®^-treated group (*p* = 0.0235). The infections in the control group were caused by the following pathogens: Methicillin-Susceptible *Staphylococcus Aureus* (MSSA) (three cases), *Staphylococcus epidermidis* (one case), and Methicillin-Resistant *Staphylococcus Aureus* (MRSA) (one case). The onset of infection was recorded at 54 ± 14 days after surgery, and there was a correlation between the type of comorbidity and the onset of infection: 4 out of 13 patients with diabetes had infection. Nevertheless, we have not described the level of glucose controlled preoperatively in this subgroup of patients, and this could limit our consideration of the correlation between diabetes and periprosthetic infection, which should instead be demonstrated through a close monitoring of plasma glucose level in the preoperative period. The other secondary endpoint at the 3-month follow-up showed no differences concerning the healing of surgical wounds between the two groups. Two patients in the control group were treated with DAIR, while five patients in the same group received prolonged antibiotic therapy, but none of the patients in the DAC^®^ group received prolonged antibiotic therapy (*p* = 0.0235). At the 6-month follow-up, no surgical wound complications were observed in any group; however, three patients who developed infection in the control group underwent hip implant revision.

No adverse events related to the impairment of the implant osseointegration process associated with the use of the DAC^®^ gel device, as evaluated by Rx-images, were observed. The results are summarized in Table 2.

## 4. Discussion

Although they have a relatively low incidence, PJIs cause severe consequences with a noteworthy deterioration in the patient’s quality of life and a significant socio-economic impact derived from the increased costs associated with managing this disease by the National Health Service and the loss of work days [45]. According to the pathophysiological mechanism of PJIs, bacteria present before and in the early stage of surgery, immediately after implant placement, “win the race for the surface”, and lay the groundwork for biofilm development and therefore infection onset [46,47]. Biofilm, a highly organized and mucopolysaccharide-based structure, particularly that formed by multidrug-resistant bacteria, represents the first step of infection persistence: when it is mature, it is difficult to eradicate from implant surfaces and surrounding tissues by agents, even when systemic antibiotic treatment combined with direct lavage and mechanical brushing is used [48]. Biofilm protects bacteria in a sessile form through the action of the immune system and antibacterial agents; therefore, through a multidisciplinary approach, it is imperative to address all efforts to prevent bacterial adhesion, especially in the early stages of surgery. Furthermore, some considerations must be given to factors that may increase the risk of contracting a PJI: there are some health-related risk factors associated with the development of PJI following total joint arthroplasty (TJA); among these, obesity, diabetes, smoking, autoimmune diseases (such as rheumatoid arthritis), and previous joint infections are considered among the main comorbidities causing a greater risk of infection along with surgical-related factors such as longer operative time and intrinsic factors like immunosuppression and malnutrition. Since TJA is an elective surgery, this allows orthopedic surgeons to clinically optimize patients before surgery to minimize their risk of developing a postoperative infection [49]. Likewise, arthroplasty revision surgery, including either septic or aseptic revision, represents an additional increased risk factor for recurrent or new PJI, compared to primary arthroplasty [50].

Also, environmental risk factors for PJIs in the operating room have to be considered as potential risk factors, and they include airborne bacteria and pathogens spread by staff and patients, operating room ventilation systems, and potentially high humidity and temperatures [51]. In conclusion, all measures adopted to reduce the risk are not always effective, and current data reported in literature are found in retrospective or prospective cohort studies [52] and need to be confirmed by larger cohorts and randomized controlled trials; therefore, complementary preventive treatments that can help to significantly reduce the risk of infection are required.

Recently, the use of hydrogels and biopolymeric-based coatings to passively protect orthopedic implants has become attractive and popular in orthopedics [53,54], and this approach has proven both effective and economically relevant [55], especially in high-risk patients affected by comorbidities. In addition to acting as a protective barrier, hydrophilic gels modify the implant surface from hydrophobic to hydrophilic by preventing the adhesion molecules present on the surface of the bacterial membrane from binding to the metal surface of the implant [56,57]. Therefore, the outcomes of this 1:1-matched case–control retrospective study support the preventive use of DAC^®^, a hyaluronan-derivative rapidly resorbable viscous hydrogel, as a temporary barrier at the hip and knee arthroplasty implant–bone interface to avoid bacterial adhesion and prevent biofilm formation and PJIs in a cohort of patients with comorbidities and risk factors. The DAC^®^ gel has been successfully used in the past, even when loaded with antibiotics, in different skeletal system surgeries, including arthroplasty, trauma, spine surgery, and oncologic orthopedics [36,41,58,59], but the primary barrier action had never been clinically demonstrated using the device alone before our clinical investigation. The DAC^®^ gel demonstrated a high safety and tolerability profile because of its natural origin (HY and PDLLA are components of connective tissue of the human body), and it did not trigger any adverse events or interfere with implant osseointegration because it acts as a temporary barrier in the early phase of surgery and immediately after implant placement, protecting the implant surfaces from bacterial colonization and being reabsorbed after 48–72 h. In fact, when considering secondary endpoints, three implant revisions at the 6-month follow-up were observed in the control group but not in the DAC^®^ group. We recognize several limitations in our study: it is a retrospective trial without randomization, with a limited sample size, and even though the PJIs observed in the control group were early infections that occurred six months after surgery, it has a relatively short-term follow-up; nonetheless, the clinical outcomes observed in our case–control cohort comparison are promising. DAC^®^ gel is user-friendly and may be indicated particularly for patients with comorbidities considered risk factors for the onset of PJIs, and may be considered in the guidelines as an additional complementary tool for the prevention of PJIs.

## 5. Conclusions

This is the first retrospective investigation that assesses the effectiveness of DAC^®^ gel used alone, without antibiotics, as a protective barrier against bacterial adhesion in a selected cohort of patients with comorbidities who have undergone arthroplasty revision and therefore are at increased risk of contracting a periprosthetic infection [7,50].

Although these clinical findings are preliminary, it was demonstrated that DAC^®^ acts as a temporary barrier, preventing bacterial colonization on the surface of the implants in the initial phase of surgery without triggering any adverse event and without interfering with the osseointegration of the implant or the healing of the surgical wound. Therefore, the device may be considered a valuable preventive point of care in arthroplasty and traumatology and is useful for reducing a phenomenon that often causes further and prolonged complications, including significant worsening of the patient’s quality of life and increases in direct and indirect costs for the National Health Service.

## Figures and Tables

**Figure 1 biomedicines-13-02408-f001:**
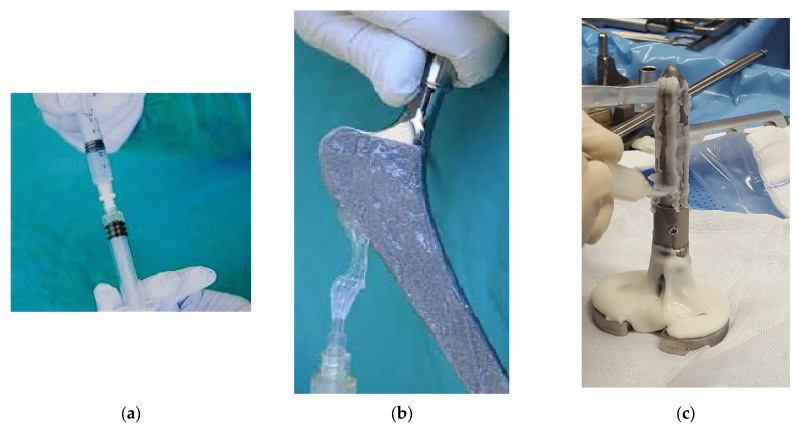
(**a**) DAC^®^ powder is hydrated with sterile water; (**b**) the hydrogel gel is uniformly spread over the hip arthroplasty implant; (**c**) the hydrogel gel is uniformly spread over the knee arthroplasty implant.

**Table 1 biomedicines-13-02408-t001:** Demographic characteristics and pre- and intraoperative data of the DAC^®^ and control groups.

Variables	DAC^®^ Group *n* = 96	Control Group *n* = 96	*p*-Value
Basic data			
Age, years, median (IQR)	63 (12.5)	63 (9)	0.7898 ⴕ
Gender, *n* male/female	82/68	82/68	1 *
Body mass index (kg/m^2^), median (IQR)	25.6 (3.3)	25.7 (3.1)	0.7307 ⴕ
Prevalent comorbidities *n* (%)			
Smoke	19 (19.8%)	8 (8.3%)	0.0224 *
Diabetes	18 (18.8%)	13 (13.5%)	0.3268 *
Obesity	16 (16.7%)	10 (10.4%)	0.2057 *
Cardiovascular disease	10 (10.4%)	10(10.4%)	1 *
Hypertension	7 (7.3%)	7 (7.3%)	1 *
CODP	7 (7.3%)	9 (9.4%)	0.6015 *
Risk factor, mean (SD)	3.8 ± 2.3	4.5 ± 3.1	0.0768 ⴕⴕ
Revision *n* (%)			
Hip	78 (81.3%)	78 (81.3%)	1 *
Knee	18 (18.8%)	18 (18.8%)	1 *
Septic	5 (5.2%)	4 (4.2%)	0.7328 *
Aseptic	91 (94.7%)	92 (95.8%)	0.7328 *
Length of surgery (min.), median (IQR)	80 (20)	90 (15)	0.0001 ⴕ

ⴕ Mann–Whitney U test; ⴕⴕ Student’s *t*-test; * chi-square test.

**Table 2 biomedicines-13-02408-t002:** Results at the 3- and 6-month follow-ups.

Variables	DAC^®^ Group *n* = 96	Control Group *n* = 96	*p*-Value
3-month follow-up			
Follow-up (days), median (IQR)	95 (12)	105 (4)	0.0001 ⴕ
Infection *n* (%)	0	5 (5.2%)	0.0235 *
Pathogen			
Methicillin-Susceptible *S. aureus* (MSSA)	-	3	
S. Epidermidis	-	1	
Methicillin-Resistant *S. Aureus* (MRSA)	-	1	
Onset of infection (days), mean (SD)		54 ± 14	
Surgical wound			
Regular	92 (95.8%)	89 (93%)	0.3515 *
Dehiscence	0	3 (3.1%)	0.0809 *
Exudate	1 (1%)	0	0.3160 *
Swelling	0	2 (2.1%)	0.1551 *
Redness	2 (2.1%)	1 (1%)	0.5606 *
Delayed wound healing	0	1 (1.3%)	0.3160 *
Requiring advanced wound medication	1 (1%)	0	0.3160 *
Other complications, *n* (%)			
DAIR	0	2 (2.1%)	0.1551 *
Prolonged antibiotic therapy	0	5 (5.2%)	0.0235 *
6 m follow-up			
Follow-up (days), median (IQR)	182.5 (14)	192 (3.5)	0.0001 ⴕ
Surgical wound			
Regular	96 (100%)	96 (100%)	1 *
Other complications, *n* (%)			
Revision (implant removal)	0	3 (3.1%)	0.0809 *

ⴕ Mann–Whitney U test; * chi-square test.

## Data Availability

All the data generated or analyzed during this study are included in this published article.

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
