# Peer review of "DAC®, a Hyaluronan Derivative in the Form of a Gel, Is Effective in Preventing Periprosthetic Joint Infection During Arthroplasty Revision in Patients with Comorbidities: A Retrospective, Observational, 1:1-Matched Case–Control Clinical Investigation"

_biomedicines, 2025, doi:10.3390/biomedicines13102408_

Round 1

Reviewer 1 Report

Comments and Suggestions for Authors

Introduction: should have more data from ECDC reports. I would also like a chapter on prosthesis material and factors having an impact on biofilm formation, if the main idea is to decrease adhesion.

I would rather suggest resigning from citation 29 due to the opposite recommendation to local antibiotic use according to the WHO and SHEA guidelines to prevent SSI. In my opinion, there is no space in the introduction to discuss this topic.

Materials and methods

Which SSI were included? Standard Infection Prevention Control works based on ECDC/CDC data, dividing them for superficial (the most commonly omitted in statistics - that's why in official reports incidence is too low), deep and organ/space.

Statistical analysis: Were all the continuous data normally distributed? If not, median and IQR should be used for description - my experience is that it is very rare that all have a normal distribution, was it checked properly?

Results: if such divided MSSA and MRSA not s. aureus and MRSA

How was the SSI classified? Deep/supperficial etc

If DM patients were so prevalent, could you assess the perioperative glucose control level?

What does that mean that in some cases there was no regular course of surgical wound - it may be one of the superficial SSI criteria

Any standard antibiotic in DAC group? 

No data about antibiotic prophylaxis - do all patients get it?

It's good to analyse the length of hospital stay in both groups.

Why follow up lower threshold below 90 days? - It's standard for SSI observation; there needs to be an excuse for that

In the DAC group, there were some infection signs - how was the SSI excluded?

Discussion:

It's extremely short compared with the introduction. You get more than 5% prevalence of SSI in the control group - you need to discuss additional factors which can have an impact on these results, because it will also have an impact on the comparison with DAC.

All factors with potential influence for SSI should be discussed, and also some assumptions should be made based on current literature due to the retrospective nature of the data.

Author Response

Dear Sir,

We would like to thank you for tacking the time to review our manuscript, and we fully agree with your comments/obervations. We have replied to each of your observations as detailed in the attached file. The corresponding revisions/corrections are highlighted in green in track changes in the re-submitted revised manuscript file.

We hope that the revisions will be favorably received.

Best regards

Giuseppe Ricciardi

Reviewer 2 Report

Comments and Suggestions for Authors

Thank you for your submission. I have two major concerns. First, the study group size is too small to support your conclusions. I suggest that a power analysis would yield a study group size well over 96. If I am incorrect, please add a power analysis to your manuscript. Next, as you know, osseointegration is a key step for prosthesis stability and longevity. I am concerned that the gel interferes with this step, and although it may prevent infections and their associated revision surgeries, A lack of osseointegration will lead to painful loosening, subsidence, and also revision surgeries. Therefore, I suggest addressing this topic directly in your discussion and conclusion sections. You are making progress with your work; however, in my opinion, we do not know enough about it yet to report that it is safe and effective. By the way, safety was not one of your study metrics, therefore it cannot be in the title in my opinion. Thankyou.

Comments on the Quality of English Language

Lines 117 and 228 need a grammar revision.

Reviewer 3 Report

Comments and Suggestions for Authors
  1. the conclusion in the abstract part( line 31-35) is weird, please check it. it is an illustration that how to write conclusion.
  2. although the authors presented positive result of decreasing the possibility of PJI with DAC® gel, it was univariate analysis study. therefore, multivariate analysis are strongly suggested.
  3. PJI mostly occurred in the joint, then extend to the medullary canal, how did you put gel in the joint after revision surgery?

Reviewer 4 Report

Comments and Suggestions for Authors

The manuscript reports an observational case-control study evaluating a cohort of patients treated in routine clinical practice who underwent hip and knee arthroplasty revision. These patients had comorbidities that could increase the risk of periprosthetic joint infections (PJIs). The study compares patients receiving an early coated implant with DAC® gel—applied immediately before placement in the anatomical site to provide a protective barrier against bacterial adhesion—to a control cohort treated with conventional procedures. The results suggest a reduction in infection rates with this approach, which appears to be of clinical benefit to patients.

However, several issues should be addressed to strengthen the manuscript:

  • Adjustment for confounding factors is needed, particularly the duration of surgery, which was longer in the control group and is a known risk factor for infection.
  • The quality of the images should be improved for better clarity and interpretation.
  • Provide long-term follow-up results, including the number of patients requiring further revision.
  • Clarify the criteria used to determine whether DAC was applied or not, since patients in the cohort were treated during the same period. This point is crucial to reduce selection bias.
  • The conclusion section should be revised. It currently reads as a summary or abstract of the paper. Instead, it should focus only on the main results, limitations of the study, and future perspectives, while removing the objective and methods.

Author Response

Dear Sir,

Thank you very much for tacking the time to review our manuscript and we fully agree with your comments/obervtions. We have replied to each of your comments that you will find in the attached file.

We have also edited the manuscript based on your comments and the corresponding revisions/corrections are highlighted in blue in track changed in the re-submitted manuscript file.

We hope that the revisions will be favorably received.

Best regards

Giuseppe Ricciardi 

Round 2

Reviewer 1 Report

Comments and Suggestions for Authors

Dear authors, I believe that the text has definitely improved in quality, but it requires some additional improvements.

Materials and methods:

1. Exclusion criteria should exclude those included, so if you are including adults, there is no underage to exclude, as well as a cement issue.

Results:

1. If not a normal distribution, use not the average, but the median age / BMI etc.

2. Text shouldn't duplicate data from the table - in the current form, there is too much of the same information.

3.  The problem of determining the median length of hospital stay and statistical significance for such a number of patients is debatable - I leave this to the editor's decision

Discussion:

1. You are pointing out in the results that DM may influence the risk because all of the SSI patients had DM. But you don't discuss it at all - add this, include limitations that you don't know the glucose control level - this is one of the leading WHO/SHEA recommendations that all patients, not only DM, should have intensive glucose control in the peri-operative period. 

Author Response

Dear reviewer 1,

Thank you to reviewing our manuscript entitled: "DAC a hyaluronan derivative in the form of a gel is effective in preventing peroprosthetic joint infection during arthroplasty revision in patients with comorbidities: a retrospective observational, 1:1 matched case-control clinical investigation." We would like to inform that we have answered to all your comments/observations, that we agree with, and which you will find in the attached file "DAC arthroplasty Response to the Reviewer 1 Comments".

We hope that our responses will be favorably accepted and that the manuscript can be published.

Best Regards 
